# The Diverse Mycorrizal Morphology of *Rhododendron dauricum*, the Fungal Communities Structure and Dynamics from the Mycorrhizosphere

**DOI:** 10.3390/jof10010065

**Published:** 2024-01-14

**Authors:** Jin Liu, Yang Xu, Yan-Ji Si, Bin-Qi Li, Peng Chen, Ling-Ling Wu, Pu Guo, Rui-Qing Ji

**Affiliations:** Engineering Research Center of Edible and Medicinal Fungi, Ministry of Education, Jilin Agricultural University, Changchun 130118, China; m17633034156@163.com (J.L.); xuyang@shbio.com (Y.X.); siyanjisyj@126.com (Y.-J.S.); l1079860934@163.com (B.-Q.L.); 18943095195@163.com (P.C.); 17390928045@163.com (L.-L.W.); guopu1102@163.com (P.G.)

**Keywords:** mycorrhiza, ericoid mycorrhiza, fungi community, ecological niches, forest type

## Abstract

It is generally believed that mycorrhiza is a microecosystem composed of mycorrhizal fungi, host plants and other microscopic organisms. The mycorrhiza of *Rhododendron dauricum* is more complex and the diverse morphology of our investigated results displays both typical ericoid mycorrhizal characteristics and ectomycorrhizal traits. The characteristics of ectendoomycorrhiza, where mycelial invade from the outside into the root cells, have also been observed. In order to further clarify the mycorrhizal fungi members and other fungal communities of *R. dauricum* mycorrhiza, and explore the effects of vegetation and soil biological factors on their community structure, we selected two woodlands in the northeast of China as samples—one is a mixed forest of *R. dauricum* and *Quercus mongolica*, and the other a mixed forest of *R. dauricum*, *Q. mongolica*, and *Pinus densiflor*. The sampling time was during the local growing season, from June to September. High-throughput sequencing yielded a total of 3020 fungal amplicon sequence variants (ASVs), which were based on sequencing of the internal transcribed spacer ribosomal RNA (ITS rRNA) via the Illumina NovaSeq platform. In the different habitats of *R. dauricum*, there are differences in the diversity of fungi obtained from mycorrhizal niches, and specifically the mycorrhizal fungal community structure in the complex vegetation of mixed forests, where *R. dauricum* is found, exhibits greater stability, with relatively minor changes over time. Soil fungi are identified as the primary source of fungi within the mycorrhizal niche, and the abundance of mycorrhizal fungi from mycorrhizal niches in *R. dauricum* is significantly influenced by soil pH, organic matter, and available nitrogen. The relationship between soil fungi and mycorrhizal fungi from mycorrhizal niches is simultaneously found to be intricate, while the genus *Hydnellum* emerges as a central genus among mycorrhizal fungi from mycorrhizal niches. However, there is currently a substantial gap in the foundational research of this genus, including the fact that mycorrhizal fungi from mycorrhizal niches have, compared to fungi present in the soil, proven to be more sensitive to changes in soil moisture.

## 1. Introduction

*R. dauricum* L., which belongs to the Ericaceae and *Rhododendron*, and is commonly known as ‘Yingshanhong’, is a second-level protected wild plant in China that is mainly distributed in northeastern China, Mongolia, the Korean Peninsula, and the Russian Far East [1]. Its significant horticultural and medicinal value [2,3,4,5]. *R. dauricum* is closely associated with its mycorrhizal microbiota, and this symbiotic relationship plays a crucial role in plant growth, adaptation, and ecosystem functionality [6,7,8]. The symbiotic partnership supports plants through nutritional supply, stress resistance, and disease protection [7,9]. 

Understanding the characteristics and interactions of mycorrhizal and soil fungi community in *R. dauricum* habitats is therefore essential for a deeper comprehension of the ecological functions, stability, and species conservation linked to this symbiotic system.

*Rhododendron* is a mycorrhizal plant, and within the family Ericaceae, all plants, with the exception of the genera *Arbutus* and *Arctostaphylos*, have a distinct type of mycorrhizal symbiosis. Fungal hyphae penetrate the root cells of Ericaceae plants to form a unique structure known as a “coil”, which Harley J. L. (1969) termed “Ericoid Mycorrhiza (ERM)” [10], and is also referred to as “ericoid mycorrhizal association”. Numerous studies now indicate the crucial role of ERM in stress physiology, ecological adaptation, nutrient uptake, and transport in Ericaceae plants [11,12]. The diversity and functional aspects of ERM fungal communities have gained extensive attention [13,14]. Furthermore, artificial inoculation with mycorrhizal fungi can, in Ericaceae plants like *Rhododendron* and *Vaccinium*, enhance stress resistance, growth, and other traits [15,16], emphasizing the practical significance of ERM mycorrhizal research. Mycorrhizae are symbiotic structures formed by host plants and mycorrhizal fungi that, in essence, constitute a microecosystem composed of mycorrhizal fungi, hyphosphera bacteria, plant root tips, soil, and other components.

During mycorrhizal symbiosis, plants establish complex symbiotic networks with mycorrhizal fungi and bacteria, achieving synergistic effects through material exchange [17] and signaling communication [18]. The microbial diversity in mycorrhizal symbiosis may enhance functional diversity among symbiotic partners [19], providing broader nutritional and environmental adaptability. Additionally, the microbial diversity within mycorrhizal symbiosis can impact plant ecosystem functionality [20], including soil carbon cycling, nutrient utilization, and disease resistance [21]. Soil structure also significantly influences the structure of fungal communities [22], thereby impacting the formation of mycorrhizae. A thorough investigation of mycorrhizal fungi diversity in plants of the Ericaceae family is therefore crucial for understanding the ecological functions and evolutionary mechanisms of their symbiotic relationships with plants.

The formation of mycorrhizal microecology is closely related to the habitat, as climatic elements [23] and edaphic features [24] in the habitat both influence the mycorrhizal microecology. Soil fungi community is a vital component of soil that has a critical influence on soil health and ecological functionality [25], and is also the direct interface with mycorrhizal interactions. The diversity of soil fungi is closely linked to their functional diversity, as different fungi communities play distinct roles in soil biotransformation and ecological processes [26]. Soil fungi contribute significantly to soil structure and functionality by driving the organic matter decomposition, nutrient cycling, and pathogen inhibition [27] and secrete binding substances that promote soil aggregation [28]. A comprehensive study of soil fungi diversity in plants of the Ericaceae family is therefore essential for revealing their interactions and functional mechanisms within soil ecosystems.

Currently, most research of *R. dauricum* focuses on its extractive value, and its habitat fungi community has only been studied to a limited extent, with the result that the characteristics of fungi community in *R. dauricum* habitats remain unclear. This study therefore employs molecular biology techniques and high-throughput sequencing methods to analyze the structure of mycorrhizal and soil fungi community in two *R. dauricum* habitats during June–July and August–September, with the aim of revealing the diversity composition and structural features of these fungal communities. Furthermore, we will assess the impact of different environmental factors on mycorrhizal and soil fungi diversity, exploring the mechanisms of interaction between mycorrhizal and soil fungi community. Through these investigations, we will gain deeper insights into the ecological functions and relationships of mycorrhizal and soil fungi in plants of the Ericaceae family, along with strategies for biodiversity conservation.

## 2. Materials and Methods

### 2.1. Sampling Locations

The sampling locations in this study are all located in the northeastern region of China: Sampling point A is located in Baoqing County, Heilongjiang Province (SA, 132°2′36.82″ E, 46°8′26.24″ N, elevation 280 m), in a mixed forest of *R. dauricum* and *Quercus mongolica* (Figure 1a,b). This sampling point has a temperate monsoon climate and is situated in a natural forest area, with volcanic ash soil. Sampling point B is located in the National Matsutake Nature Reserve in Longjing City, JiLin Province, China. (SB, 129°40′16.65″ E, 42°33′39.60″ N, elevation 380 m), in a mixed forest of *R. dauricum* with *Q. mongolica*, and *P. densiflor* (Figure 1c,d). This sampling point has a temperate monsoon climate and is situated in a natural forest area, with volcanic ash soil. Both locations are devoid of any other woody plants.

### 2.2. Collection and Preparation of Mycorrhiza Root Tips’ and Soil Samples

In the months of June to July and August to September, two sampling points were selected within a 100 m × 100 m area. The five-point sampling method was followed to collect samples within each plot [29]. Mycorrhizal root tips samples were collected from areas with abundant fine roots approximately 0.8 m (depth of 5–15 cm) away from the tree trunk in four cardinal directions and the central area. The surface humus and soil layers were removed using a shovel, and fine root tips (approximately 50 g) were taken. The collected fine root tips were placed in self-sealing bags and combined to form the mycorrhiza root tip samples. The sample was transported back to the laboratory using dry ice for preservation.

The mycorrhiza root tip samples were processed by being treated with a 0.1% Tween 20 solution for 60 min and then rinsed with sterile water. After excess moisture was removed, the mycorrhiza root tip samples (with enlarged root tips) were selected and placed in centrifuge tubes for storage at −70 °C.

Soil samples within a 10 cm radius around the mycorrhiza root tip sample points were placed in self-sealing bags and transported back to the laboratory using dry ice, and then stored at −80 °C for further analysis.

These procedures ensured proper collection, handling, and preservation of mycorrhizal and soil samples for subsequent laboratory analyses.

### 2.3. Collection and Processing of R. dauricum Mycorrhizal Root Tips Samples

Collect small and fine roots of *R. dauricum*, ranging from 5 cm to 30 cm (taking into account the depth of root distribution), from the soil layer. Place the roots in soil bags, attach labels, and record the collector’s name, sampling time, location, plant name, and surrounding environmental information. In the understory, the root systems of different plants intermingle so, when taking root samples, dig along the primary root of the plant to ensure that the collected roots are indeed connected to the main root of the selected plant and try to take root segments with root tips whenever possible.

Fix the root samples in a 1/2 FAA fixing solution (5 mL formalin, 5 mL glacial acetic acid, 90 mL 70% alcohol) for mycorrhizal morphology observation experiments. Rinse the root tips samples fixed in FAA fixing solution with distilled water 2–3 times, and cut them into approximately 1 cm-long root segments. Decolorize the samples in a 10% KOH solution at 92 °C for 20–60 min and then rinse them thoroughly with distilled water. Stain the samples with a 0.05% trypan blue glycerol solution for 2–4 h, differentiate with glycerol, and prepare slides [30]. Observe the mycorrhizal structure and take photographs using a Zeiss Axio Scope A1 microscope (Carl Zeiss, Jena, Germany). Determine the mycorrhizal morphological types based on the research by Genre et al. [31] (Appendix A).

### 2.4. DNA Extraction and Illumina NovaSeq Platform

Following the manufacturer’s instructions, the DNeasy PowerSoil Kit (QIAGEN, Inc., Hilden, Germany) was used for DNA extraction. The extracted DNA samples were stored at −20 °C before data analysis. The quantity and quality of the extracted DNA samples were measured by using the NanoDrop ND-1000 spectrophotometer (Thermo Fisher Scientific, Waltham, MA, USA) and agarose gel electrophoresis, respectively.

For fungal ITS1 region PCR amplification, the primers ITS5 and ITS2 were used [32]. The PCR amplicons were purified using Agencourt AMPure Beads (Beckman Coulter, Indianapolis, IN, USA) and quantified using the PicoGreen dsDNA Assay Kit (Invitrogen, Carlsbad, CA, USA). After individual quantification, the amplicons were pooled in equal amounts.

The pooled amplicons were subjected to paired-end 2 × 300 bp sequencing using the Illumina NovaSeq platform provided by Shanghai Parsenon Bioinformatics Co., Ltd. (Shanghai, China).

### 2.5. Soil Physical and Chemical Properties Analysis

*Drying the Soil Samples*: dry the soil samples using silica gel beads for approximately 1–2 weeks until they are completely dried. *Sieve Separation*: use a 2 mm sieve to remove stone particles from the soil samples. *Available Phosphorus (AP) Determination*: measure the soil’s available phosphorus using a 0.5 mol·L^−1^ NaHCO_3_ solution and the molybdenum blue colorimetric method [33]. *Soil Organic Matter (SOM) Determination*: determine the soil’s organic matter content using the potassium dichromate volumetric method [34]. *Available Nitrogen (TN) Determination*: measure the soil’s available nitrogen (hydrolyzable nitrogen) using the alkaline diffusion method [35]. *Available Potassium (AK) Determination*: determine the soil’s available potassium using a 1 mol·L^−1^ NH_4_OAc extraction and flame photometry [36]. *Soil pH Measurement*: measure the soil pH using the potentiometric method: the ratio of water to soil is 2.5:1, with an oscillating time of 2 min, followed by a settling period of 30 min [37].

*Determining soil Texture based on USDA-NRCS Soil Texture Classes Standards* [38]: first, remove SOM (Soil Organic Matter) and carbonate by adding 10% hydrogen peroxide (GR) and 0.2% hydrochloric acid (GR) to 10 g of sieved soil (2 mm sieve) in a 250 mL beaker. Then add 10 mL of 0.05 mol·L^−1^ sodium hydroxide solution (GR) as a dispersant and shake the beaker uniformly before analysis. Finally, wash the soil sample with water in a settling cylinder. Determine the soil composition by using the pipette method. Soil texture type is determined on the basis of the content of clay (<0.002 mm particle size), silt (0.02–0.0002 mm particle size), and sand (0.02–2 mm particle size).

### 2.6. Climate Elements Collection

The climate elements data—air average temperature (AT), monthly precipitation (MP), surface average temperature (ST) and relative humidity (RH)—were downloaded from the China meteorological data network [39].

### 2.7. Bioinformatics

Microbiome bioinformatics analysis was performed using QIIME2 2019.4 [40], with slight modifications based on the referenced literature. The demux plugin was used to demultiplex the raw sequence data, followed by primer trimming using the cutadapt plugin [41]. The DADA2 plugin was employed for quality filtering, denoising, merging, and the removal of chimeras from the sequences [42]. The non-singleton Amplicon Sequence Variants (ASVs) were aligned with MAFFT [43] and used for phylogenetic tree construction using FastTree2 [44]. The ASVs were annotated using the naive Bayes classifier technique in the feature-classifier plugin [45], based on the SILVA Release 132/UNITE Release 8.0 database [46].

### 2.8. Data Processing

The data were processed using SPSS 19.0 software for one-way analysis of variance (ANOVA) to analyze differences in soil physical and chemical properties and climate elements among different samples. All statistical analyses were evaluated at α = 0.05.

RDA (Redundancy Analysis) was employed to analyze the relationship between fungal composition and soil factors and climate elements. Network analysis is based on the Spearman correlation results, involving the calculation of correlation coefficients between feature values to identify connections between variables. This method utilizes network graphs or connection models to illustrate the internal structure of data. The significant correlations between feature nodes are represented by connecting lines, providing a visual representation of the interactions between different species. One-way analysis of variance and Tukey’s HSD test at a significance level of 5% were used to test the effects of forest development on soil properties and fungal characteristics. R version 3.1.1 was used for visualizing RDA and ANOVA.

### 2.9. Fungal Community

Samples were collected from different forest types and at different sampling times to explore the influence of these habitats and periods on the mycorrhizal fungal community. β-diversity distance was used to describe the differences between samples, and so was NMDS analysis, which arranges the distances between fungal community samples by calculating the Bray-Curtis dissimilarity matrix at the ASV level. NMDS analysis is not influenced by the numerical values of sample distances, but only considers the relative distances between samples, and its results are more reliable when the stress value is less than 0.2. 

We used the Chao 1 index, Simpson index and Shannon index [47] to assess fungal α-diversity—a higher Simpson index indicates greater community diversity. We created a Venn diagram using the ASV abundance table to describe the differences between ASVs among samples. The genera with ASV relative abundance (≥1.00%) are considered dominant [48]. 

## 3. Results

### 3.1. Mycorrhizal Ecological Niches of R. dauricum Habitats

#### 3.1.1. *R. dauricum* Mycorrhizal Morphology

The overall morphology of mycorrhizal fungi is shown in Figure 2a. Through the observation of mycorrhizal anatomical morphology, various mycorrhizal structures were observed, with mycelia distributed within root cells (Figure 2b) (ERM) and the root surface (Figure 2c), and also seen to be invading root cells from the outside (Figure 2d) (AM). The observed hyphae had branches and septa. Mycelial structures within the cells exhibited significant variation, including intracellular mycelial coil, intracellular dissolving mycelium, and penetrating mycelium. Mycelia between cells mostly grew longitudinally along the root. There was a significant amount of mycelium on the root surface, with some mycelia forming a mantle on the root surface. Most mycelia however traveled along the outer root wall.

#### 3.1.2. Diversity of Fungi and Mycorrhizal Fungi in Ecological Niches of *R. dauricum* Habitats

The fungal alpha diversity indices of mycorrhiza root tip samples from the same habitat at different time periods showed no significant differences (*p* > 0.05). Within each habitat, the Chao1, Shannon and Simpson indices for samples collected in June–July were higher than those for samples collected in August–September, indicating higher diversity and richness in June–July, suggesting an increase in diversity and richness in the mycorrhizal niche during this time interval. When comparing different habitats during the same period, the SB site exhibited higher average Chao1, Shannon and Simpson indices compared to the SA site, indicating higher diversity and richness in the SB site. The fungi in SB mycorrhizal samples displayed higher diversity and richness, compared to site SA (Figure 3a).

NMDS analysis was conducted by calculating the Bray-Curtis dissimilarity matrix at the ASV level to rank the distances between true community samples. The stress value was 0.000078 (<0.2), indicating the reliability of the analysis. The fungi communities from mycorrhiza root tip samples in the SA site were closer in distance, indicating higher similarity in composition structure, while those in the SB site were more distant, indicating greater compositional differences. The fungi communities from mycorrhizal samples in different habitats during the same period exhibited greater distances and compositional differences (Figure 3b). Using the UNITE Release 8.0 and Silva Release 132 database, cluster analysis was performed on fungal amplicon sequence variants (ASVs). A clustering heatmap was generated for the top 20 genera, and the fungal community was categorized into 13 phyla, 46 orders, 102 families, and 216 genera. The dominant genera were those with a relative abundance exceeding 1%. The fungal communities in different habitats exhibit significant distances and substantial structural differences. In the same period of habitat, the fungal community structures were closer in proximity, with smaller differences between fungal communities at different times in SB, compared to SA. Inter-group differences were not statistically significant (R = 1, *p* = 0.1 > 0.05). (Figure 3c). Of the top twenty genera, eight belong to ECMF (Ectomycorrhizal Fungi), six to SAF (Saprotrophic Fungi), three to ERMF (Ectomycorrhizal and Saprotrophic Fungi), one to ORMF (Orchid Mycorrhizal Fungi), and two fall into other categories (Figure 3d). SAQG had five dominant genera, which were *Russula* (22.46% ECMF), *Cortinarius* (15.81% ECMF), *Phialocephala* (5.48% ERM), *Tricholoma* (3.22% ECMF), and *Cladophialophora* (1.29% SAF). SAHG had seven dominant genera, including *Russula* (31.92% ECMF), *Tricholoma* (6.24% ECMF), *Mortierella* (5.05% other), *Trichoderma* (4.79% SAF), *Cladophialophora* (4.25% SAF), *Cortinarius* (2.47% ECMF), and *Umbelopsis* (1.94% SAF). SBQG had seven dominant genera, namely *Phialocephala* (26.05% ERM), *Lachnum* (8.75% ECMF), *Cladophialophora* (6.94% SAF), *Mortierella* (6.14% SAF), *Trichoderma* (2.85% SAF), *Russula* (1.58% ECMF), and *Umbelopsis* (1.19% SAF). SBHG had six dominant genera, including *Phialocephala* (22.71% ERM), *Trichoderma* (5.76% SAF), *Cladophialophora* (5.51% SAF), *Lachnum* (4.67% ECMF), *Hydnellum* (4.50% ECMF), and *Mortierella* (2.92% SAF) (Appendix A), with ECMF being the dominant group of genera.

The ASV Venn diagrams enable the examination of differences in fungal community species richness and composition (Figure 3e). In SA, for mycorrhizal samples collected in both June–July and August–September, there were 238 shared ASVs, with 783 and 316 unique ASVs respectively. In SB, the mycorrhizal samples from June–July and August–September had 386 shared ASVs, with 1178 and 492 unique ASVs respectively. The mycorrhizal samples from June–July in SA and SB shared 357 ASVs, with 664 and 1207 unique ASVs respectively, while the mycorrhizal samples from August–September in SA and SB shared 205 ASVs, with 349 and 673 unique ASVs respectively.

### 3.2. The Impact of Habitat Soil on the Mycorrhizal Fungal Community in the Habitat of R. dauricum

#### 3.2.1. The Relationship between Abiotic Factors and Mycorrhizal Fungi in Mycorrhizal Samples

In SA, the soil pH (pH), soil organic matter (SOM), soil available potassium (AK), soil available nitrogen (TN), and soil available phosphorus (AP) contents in July were all higher than those in August–September. Among these soil factors, except for soil available potassium, all others showed significant differences (*p* < 0.05). In SB, the soil factors in July were, except for soil organic matter content, higher than those in August–September. Soil available potassium, soil available nitrogen, and soil available phosphorus exhibited significant differences (*p* < 0.05) between the two periods (Table 1).

The relationship between mycorrhizal fungi community composition and soil factors was analyzed using RDA. The RDA axes (1 and 2) explained 75.19% and 11.26% of the variance in mycorrhizal samples, respectively (Figure 4a). The *Cortinarius* and *Mycena* exhibit a high correlation with SAQG, while AP and SOM also show a strong correlation with SAQG. *Lachnum* and *Hydnellum* demonstrate a notable correlation with SAHG, and Sand is highly correlated with SAHG. *Tricholoma* and *Pseudotomentella* show a significant correlation with SBQG. Additionally, *Lachnum* and *Phialocephala* exhibit a high correlation with SBHG, while Sand and pH also display a strong correlation with SBHG. Soil pH (*p* = 0.001), soil organic matter (*p* = 0.007), and soil available nitrogen (*p* = 0.005) significantly influenced fungal abundance in mycorrhizal (*p* < 0.05). 

Comparing SA and SB, it was observed that SA had a higher proportion of sand in the soil composition, while SB had higher proportions of silt and clay. These results indicate that both SA and SB have sandy loam soils (Figure 4b). 

#### 3.2.2. Fungal Species Composition in the Soil of *R. dauricum* Habitat

The interactive bar chart displays the top 20 fungal genera in soil from the two sites. The abundance of fungal genera in the SA fluctuated significantly over the two periods, while the SB exhibited comparatively smaller fluctuations (Figure 5). There are five dominant genera in SAQT, namely *Russula* (31.23% ECMF), *Cortinarius* (16.49% ECMF), *Umbelopsis* (3.59% SAF), *Mortierella* (2.04% SAF), and *Tricholoma* (1.33% ECMF). SAHT had five dominant genera, which were *Tricholoma* (42.35% ECMF), *Umbelopsis* (23.38% SAF), *Russula* (18.05% SAF), *Mortierella* (2.88% SAF), and *Trichoderma* (2.49% SAF). SBQT had five dominant genera, which were *Russula* (24.61% ECMF), *Mortierella* (9.36% SAF), *Umbelopsis* (6.37% SAF), *Tricholoma* (3.64% ECMF), and *Cladophialophora* (1.81% SAF). SBHT had four dominant genera, including *Russula* (40.08% ECMF), *Hydnellum* (37.43% ECMF), *Umbelopsis* (2.15% SAF), and *Cortinarius* (1.23%ECMF) (Appendix A).

#### 3.2.3. The Relationship between Fungi Community in the Soil and Mycorrhizal Fungi in Mycorrhizal Ecological Niches of the Habitat of *R. dauricum*

The network diagram reveals the mutual relationships and correlations between soil fungi and mycorrhizal fungi within the mycorrhiza root tip samples. The diagram consists of 102 nodes and 738 edges, with 126 and 612 edges displaying negative and positive correlations, respectively. Among these, mycorrhizal fungi are involved in 78 edges, with 16 and 62 edges exhibiting negative and positive correlations (Figure 6). Betweenness centrality assesses the pivotal role of nodes in facilitating communication within the network. Nodes with higher values are crucial for maintaining connections, and their absence may disrupt communication pathways. Of the top ten genera based on betweenness centrality, two (*Hydnellum* and *Pseudotomentella*) belong to mycorrhizal fungi in the mycorrhiza root tip samples, indicating their significant role in facilitating species interactions within the entire fungal community (Appendix A).

### 3.3. The Relationship between Climate Factors and Mycorrhizal Fungi in Mycorrhizal Ecological Niches of the Habitat of R. dauricum

The air average temperature (AT) in SA of the *R. dauricum* habitat for the months of June–July was higher than that in August–September, with no significant difference (*p* = 0.056 > 0.05); the monthly precipitation (MP) in August–September was significantly higher than that in June–July (*p* = 0 < 0.05); the surface average temperature (ST) in June–July was higher than that in August–September, with no significant difference (*p* = 0.255 > 0.05); the relative humidity (RH) in August–September was significantly higher than that in June–July (*p* = 0.034 < 0.05) (Table 2).

In SB, the AT in August–September was higher than that in June–July, with no significant difference (*p* = 0.056 > 0.05); the MP in June–July was significantly higher than that in August–September (*p* = 0 < 0.05); the ST in August–September was higher than that in June–July, with no significant difference (*p* = 0.255 > 0.05); the RH in August–September was significantly higher than that in June–July (*p* = 0.034 < 0.05) (Table 2).

Using RDA, the relationship between mycorrhizal fungi community composition and climate elements was analyzed. The explained variance of RDA axes 1 and 2 was 66.34% and 9.76%, respectively (Figure 7), where rainfall (*p* = 0.001) and relative humidity (*p* = 0.024) significantly influenced fungal abundance in mycorrhizal (*p* < 0.05).

## 4. Discussion

*Rhododendron* plants, which belong to the Ericaceae family, typically have fine, delicate fibrous roots that consist of a single layer of epidermal cells, one to two layers of cortical cells, and a slender stele. Mycorrhizal structures in these roots do not significantly differ from uninfected fibrous roots, and the key difference is the presence of mycelia (septate hyphae with branches) or mycelial nodes within mature epidermal and cortical cells. The mycelia in the mature epidermal region of the fibrous roots form a loose network, which then penetrates the epidermal cells, usually through a single entry point but sometimes through multiple entry points. The observed mycorrhizal morphology of *R. dauricum* is quite complex, showing both typical ericoid mycorrhiza (ERM) features (such as mycelial nodes, mycelial clusters, and mycorrhizal sheaths) as well as intracellular penetrating mycelia, mycelial networks on the root surface, wandering mycelia on the root surface, and degraded mycelia. In some root samples, typical ectomycorrhiza (ECM) features, namely intercellular mycelia, were also observed [49]. This suggests that *R. dauricum* can form both ericoid mycorrhiza and, in specific environmental conditions, ectomycorrhiza. The characteristics of ectendoomycorrhiza, where mycelial invade from the outside into the root cells, have also been observed (AM). Some studies suggest that under certain conditions, *R. dauricum* can also form other mycorrhizal associations [50], although not typically mycorrhizal types. The exact mechanisms behind this dual capability require further research and will be a focus for our future studies. 

From the anatomical examination of intracellular mycelia involved in mycorrhizal infection, we found that there is considerable diversity among the fungi forming mycorrhizal associations with *R. dauricum*. Some fungi can form intracellular mycelial clusters and grow longitudinally between cells, while others form typical mycelial nodes within cells. Sometimes, both types of mycelia coexist in the roots of the same plant. This phenomenon has been previously noted in the research of other studies [51]. Many researchers believe that the mycorrhizal fungi associated with *Rhododendron* plants are a heterogeneous group, and the exact number of species within this group is currently difficult to determine [52,53,54]. 

We conducted sampling studies in two periods, June–July and August–September, in two habitats of *R. dauricum*. Site SA is a mixed forest of *R. dauricum* and *Q. mongolica*, while site SB is a mixed forest of *R. dauricum*, *Q. mongolica*, and *P. densiflor*. Our findings indicate that the fungi from mycorrhizal samples of both sites exhibited higher Alpha diversity indices in August–September, compared to June–July, suggesting an increase in diversity and richness in the mycorrhizal niche during this time interval. Additionally, during the same period, the fungi in SB mycorrhizal samples displayed higher diversity and richness, compared to site SA. The composition of fungal communities in mycorrhizal samples varied significantly between different habitats, and the degree of community structure change differed over time. Most fungi in mycorrhizal samples were identified as ectomycorrhizal fungi.

The variation in fungal community diversity is influenced by multiple factors, such as temperature [55], moisture [56], organic matter content [57], and other abiotic factors. In this study, the fungal communities within the mycorrhizal and soil ecological niches were directly or indirectly affected by abiotic factors. Both mycorrhizal and soil fungal communities can be directly or indirectly influenced by abiotic factors [58,59,60,61,62]. Soil serves as the immediate habitat for soil microbes and, as such, its physical and chemical properties directly impact the resident microorganisms. Microbes within the mycorrhizal can be indirectly impacted by soil conditions [63]. We analyzed the relationship between mycorrhizal ecological niche fungal communities in the *R. dauricum* habitat and the soil’s physical, chemical, and climatic elements. Redundancy analysis (RDA) revealed that the soil’s physical, chemical, and climatic elements significantly influenced fungal abundance. Soil variables are intricately intertwined, collectively shaping the structure and function of fungal communities. Some studies have demonstrated that soil organic matter, soil available nitrogen, soil available phosphorus, available potassium, and soil pH all have a significant impact on fungal community structure [64]. However, in this particular study, the influence of soil available phosphorus is not significant, even when the content of the soil’s available phosphorus is lower than the levels of the soil’s available nitrogen and potassium. Different types of fungi respond differently to these variables, making the impact of soil factors on fungal communities complex [65,66]. Additionally, soil fertility plays a promoting role in the colonization of mycorrhizal fungi, such as arbuscular mycorrhizal fungi [67]. However, the effects of the same factor on a given fungal genus differed across different ecological niches [68], which suggests that factors beneficial to a species within the mycorrhizal might be negatively correlated with the same species in the soil. This observation underscores the selective absorption of various substances from the soil by plant mycorrhizal to ensure their normal growth [69,70]. Alternatively, it could be related to mycorrhizal-associate fungi that are negatively correlated with certain factors, which might produce signaling molecules through their metabolic activities [71,72,73]. These molecules could stimulate plant physiological responses, thereby counteracting the negative impact of those factors on fungal survival [74,75,76,77]. However, this hypothesis requires extensive experimentation for validation. *R. dauricum* is an acidophilic plant, implying that the soils in its habitat play a selective role for microbes. Prior to this study, we anticipated that the mycorrhizal fungi community composition in corresponding ecological niches of the two sites would be similar or closely related. However, our results contradicted this assumption and confirmed that other vegetation in the plant habitat can indeed influence the fungal communities—both soil and mycorrhizal [78,79,80]. SB (Chao1 695.189, Shannon 6.753, Simpson 0.978) exhibited higher diversity and richness compared to SA (Chao1 451.140, Shannon 5.519, Simpson 0.927), indicating that soil fungal community diversity and richness tend to increase with greater forest type complexity, a conclusion also supported by previous research [81,82]. This diversity of niches can support a variety of fungal species with different ecological roles and preferences. Ecosystems with higher plant species richness often exhibit greater resilience and stability, and this stability can extend to the fungal community, as a diverse and stable plant community can provide a more stable habitat for fungi. Greater richness of plant species is one of the reasons for the greater fungal diversity in SB, although we do not believe it is the sole factor.

Numerous studies have demonstrated the intentional recruitment of soil microorganisms by plant roots [83,84], which is a conclusion that appears applicable to our research, although not without caution, as, from the perspective of individual plants, the recruitment of soil microorganisms by plant roots should be based on their positive impact on their own growth and development. In our study, species such as *Trichoderma* [85] and *Phialocephala* [86] have been confirmed to exert beneficial effects on plants, while many others remain unclear. The composition of plant habitat microbial communities is complex [87,88], although different fungal groups can perform similar functions. This suggests that, despite the significant differences in mycorrhizal fungi communities within the same ecological niche between the two distinct *R. dauricum* habitats, their functions in influencing plant growth and development might be analogous; for example, the study by Zhu et al. [89] lends some support to this conjecture. Root exudates play a vital role in shaping soil and mycorrhizal fungal communities [90]. Substances like organic acids and sugars can promote fungal proliferation [91], and the types of exudates can also be influenced by the soil’s physical and chemical factors [92,93], which might contribute to the substantial differences observed in fungal communities between the two *R. dauricum* habitats. Fungal community research often explores their collective roles in evaluating their impact. For any living entity, the criterion for success remains consistent: namely completing its life history. The mycorrhizal symbiotic mechanism therefore represents a mutually beneficial selection between microorganisms and plants.

The relationship between the two communities is complex, as observed in the mycorrhizal network diagram within the mycorrhizal root. Different fungi within the samples interact with each other, either constraining or promoting each other. The central fungus in the mycorrhizal samples belongs to the genus *Hydnellum*, which is an important group of stalked aquatic fungi capable of forming ectomycorrhizal associations with various woody plants. Current, research of this genus remains limited, with most studies focusing on taxonomy and chemistry [94,95], and so further ecological investigations are needed to enhance our understanding of the genus.

In this study, two climatic elements (average precipitation and relative humidity) significantly influence the mycorrhizal fungi in the mycorrhizal samples. Both factors directly impact soil moisture, with some research indicating that the effect of soil moisture on soil fungi in certain plant habitats is relatively small [96]. Soil moisture affects the photosynthetic capacity of plants, thereby influencing nutrient transport within plant tissues. Since mycorrhizal fungi inhabit the root tips, soil moisture directly and indirectly affects the abundance of mycorrhizal fungi within the roots. Certain ectomycorrhizal fungi can also assist plants in enhancing their ability to withstand drought stress [97]. Moisture varies over time, and there are significant differences in mycorrhizal roots within the same habitat, indicating the sensitivity of mycorrhizal fungi to changes in moisture levels.

## 5. Conclusions

This study observed the intricate morphological characteristics of mycorrhizae in *R. dauricum*, which encompass both endomycorrhizae and ectomycorrhizae. The simultaneous presence of both endomycorrhizae and ectomycorrhizae, representing typical features of ericoid mycorrhizae (ERM), ectomycorrhizae (ECM) and arbuscular mycorrhizae (AM), were noted. High-throughput sequencing technology was employed to sequence fungal ITS1, revealing the diverse patterns and differences that exist in fungal communities, and specifically within mycorrhizal niches in *R. dauricum* habitats with different vegetation compositions.

The results showed variations in fungal diversity within mycorrhizal niches, with the stability of mycorrhizal fungal community structures increasing with the complexity of vegetation composition. The top 20 genera primarily consisted of ectomycorrhizal fungi. Soil pH, organic matter, and available nitrogen significantly influenced the abundance of mycorrhizal fungi within the mycorrhizal niches. The relationship between soil fungi and mycorrhizal fungi is complex, with soil fungi serving as a source for mycorrhizal fungi. The genus *Hydnellum* emerged as a central genus among mycorrhizal fungi, and there is accordingly currently a significant gap in fundamental research of this genus, meaning that further research is needed to understand its ecological role.

Climatic elements also significantly impacted the abundance of mycorrhizal fungi within the mycorrhizal niches, with mycorrhizal fungi within these niches proving more sensitive to changes in moisture than soil fungi. We suggest that conservation efforts for *R. dauricum* should consider collaborative efforts with other plant species to establish a stable fungal community capable of resisting adverse environmental influences.

## Figures and Tables

**Figure 1 jof-10-00065-f001:**
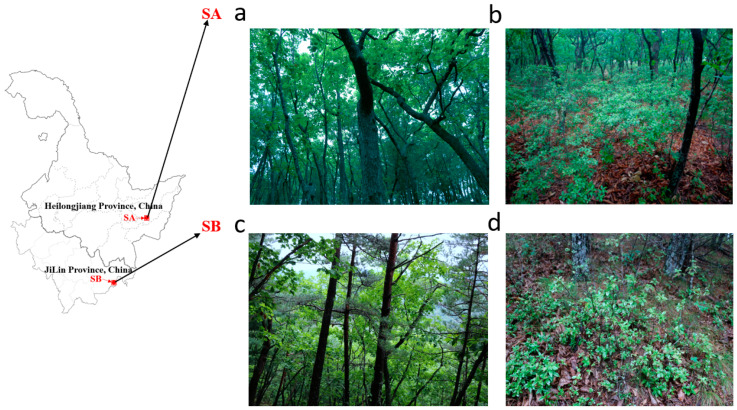
Habitat map of two sampling sites. (**a**,**b**) sampling point A (SA) is covered by *R. dauricum* mixed *Q. mongolica* forest. (**c**,**d**) sampling point B (SB) is covered by *R. dauricum* mixed *Q. mongolica* and *P. densiflora* forest.

**Figure 2 jof-10-00065-f002:**
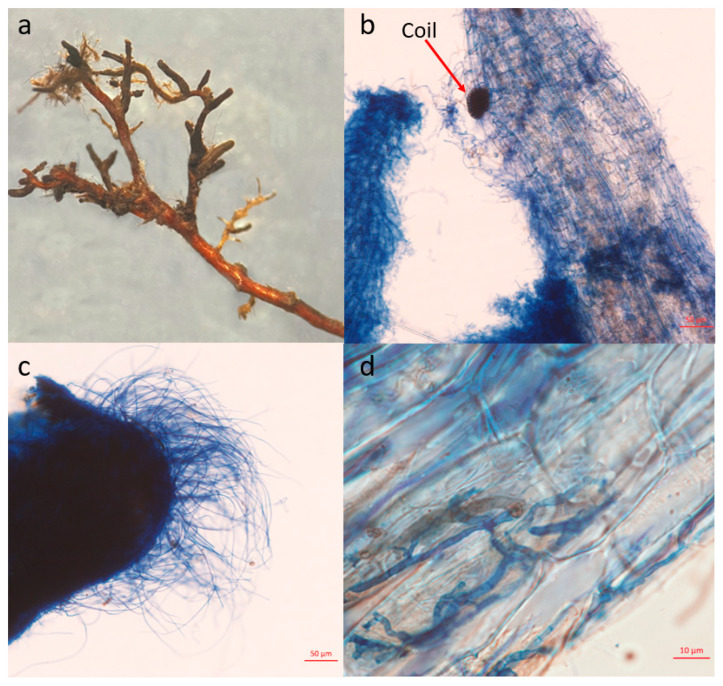
Mycorrhizal Morphology of *R. dauricum*: (**a**) Overall view of the root tips under a dissecting microscope. (**b**) Intracellular mycelial coil. (**c**) Mycorrhizal mantle. (**d**) Intracellular invading mycelium from outside.

**Figure 3 jof-10-00065-f003:**
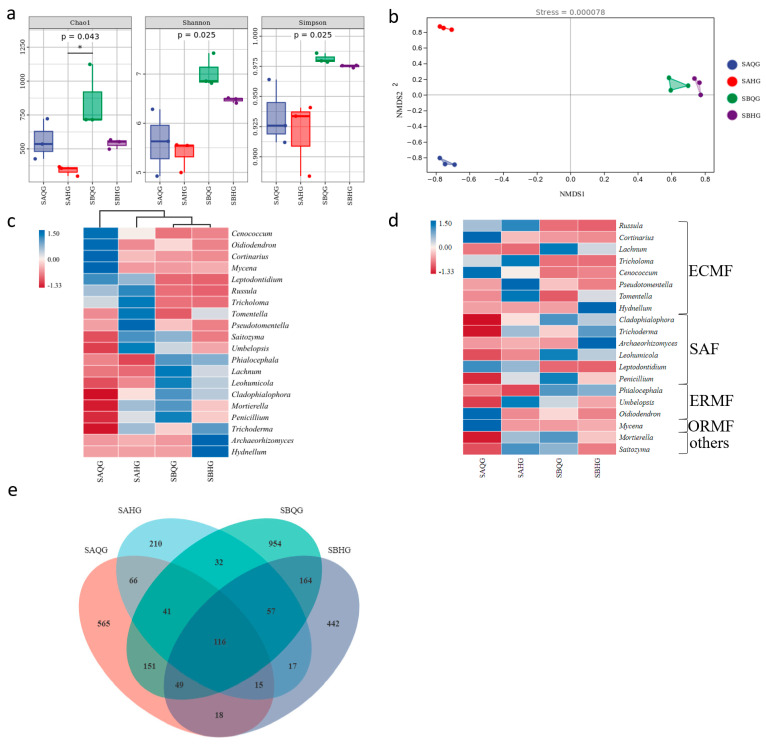
(**a**) Alpha diversity box plots of fungal community in mycorrhizal ecological niches of *R. dauricum* babitats, “*” indicates significant differences between two samples. (**b**) NMDS analysis of fungal communities in mycorrhizal ecological niches of *R. dauricum* habitats. (**c**) heat map of the fungi composition of the first 20 genera. (**d**) Functional guild cluster diagram of fungi of the first 20 genera. ECMF: ectomycorrhizal fungi; SAF, saprotrophic fungi; ERMF, ericoid mycorrhizal fungi; ORMF: Orchid mycorrhizal fungi. (**e**) Venn map of fungi from mycorrhizal ecological niches. Note: SAQG represents mycorrhizal samples from SA site in June to July, SAHG represents mycorrhizal samples from SA site in August to September; SBQG represents mycorrhizal samples from SB site in June to July, SBHG represents mycorrhizal samples from SB site in August to September, and so forth.

**Figure 4 jof-10-00065-f004:**
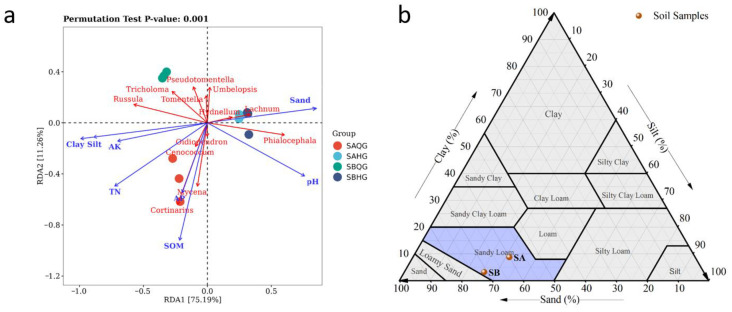
(**a**) Relationship between soil physical and chemical factors and fungal community in mycorrhiza root tip samples collected from different locations in the habitat of *R. dauricum*; (**b**) Triangular diagram (USDA) of soil texture in the habitat of *R. dauricum*.

**Figure 5 jof-10-00065-f005:**
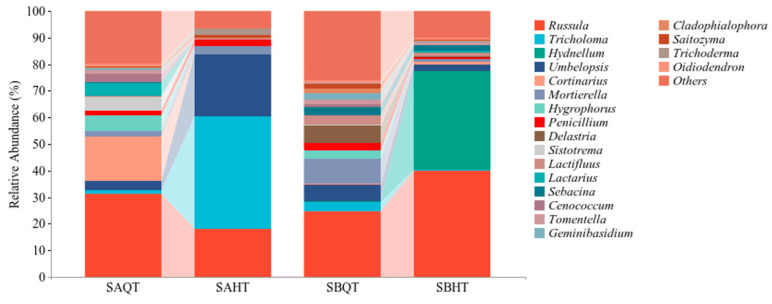
Bar chart depicting the relative abundance of fungal genera in the soil of the *R. dauricum* habitat. Note: SAQT represents soil samples from SA site in June to July; SAHT represents soil samples from SA site in August to September; SBQT represents soil samples from SB site in June to July; and SBHT represents soil samples from SB site in August to September.

**Figure 6 jof-10-00065-f006:**
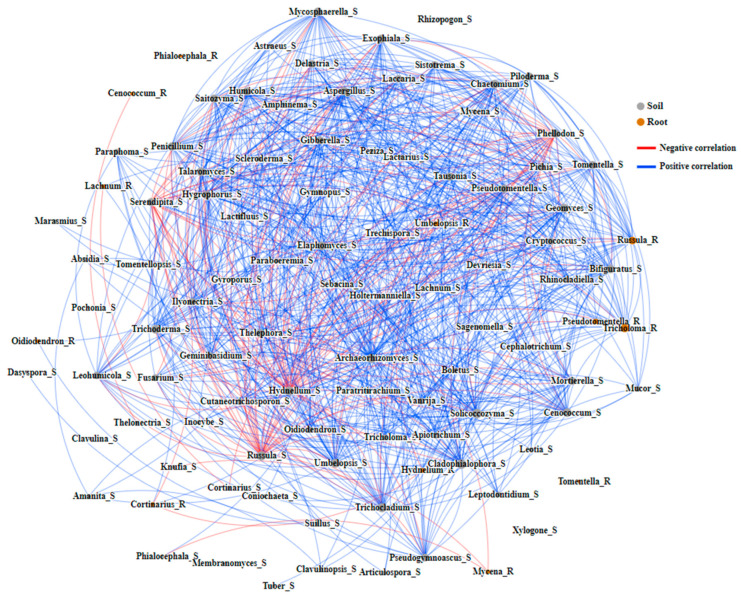
Network diagram depicting the interactions between mycorrhizal fungi from the top 20 fungal in mycorrhizal ecological niches and soil fungal community of the *R. dauricum* habitat.

**Figure 7 jof-10-00065-f007:**
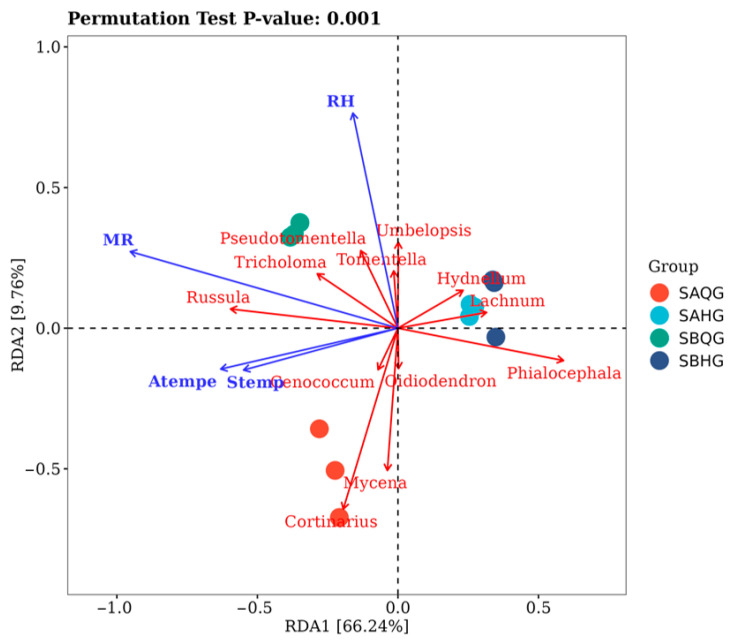
Relationships between climate elements and mycorrhizal fungi in mycorrhizal ecological niches of the habitat of *R. dauricum*.

**Table 1 jof-10-00065-t001:** One-way analysis of variance of abiotic factors in the soil.

	pH	SOM	AK	TN	AP
		(mg·kg^−1^)	(mg·kg^−1^)	(mg·kg^−1^)	(mg·kg^−1^)
F-value	12.226	18.994	26.324	17.309	10.88
*p*-value	0.002	0.001	0	0.001	0.003
SAQT	5.14 ± 0.08 ^b^	79,283.25 ± 2061.76 ^a^	26.30 ± 1.11 ^a^	122.45 ± 1.92 ^a^	7.29 ± 1.78 ^a^
SBQT	5.36 ± 0.03 ^a^	57,661.96 ± 3500.07 ^bc^	24.20 ± 1.19 ^a^	100.83 ± 0.93 ^b^	5.86 ± 0.13 ^a^
SAHT	4.86 ± 0.07 ^c^	56,806.44 ± 664.50 ^c^	25.37 ± 0.49 ^a^	106.19 ± 5.54 ^b^	2.05 ± 0.11 ^b^
SBHT	5.22 ± 0.04 ^ab^	64,972.75 ± 2430.68 ^b^	16.33 ± 0.50 ^b^	88.81 ± 3.13 ^c^	1.01 ± 0.36 ^b^

Note: The data are represented by the mean ± SD, n = 3. Different letters in the table represent a significant difference at the *p* < 0.05 level. SAQT represents soil samples from the SA site in June to July; SAHT represents soil samples from the SA site in August to September; SBQT represents soil samples from the SB site in June to July; SBHT represents soil samples from the SB site in August to September.

**Table 2 jof-10-00065-t002:** One-way analysis of climate elements in the habitat of *R. dauricum*.

	AT (°C)	MP (mm)	ST (°C)	RH (%)
F-value	3.873	53.921	1.643	4.794
*p*-value	0.056	0	0.255	0.034
SAQ	17.80 ± 1.00 ^a^	156.77 ± 20.00 ^b^	17.86 ± 2.00 ^a^	73.30 ± 1.00 ^b^
SAH	17.71 ± 0.50 ^a^	219.875 ± 10.00 ^a^	17.45 ± 1.00 ^a^	76.69 ± 1.00 ^a^
SBQ	16.12 ± 0.20 ^b^	130.46 ± 11.99 ^c^	15.26 ± 2.00 ^a^	73.87 ± 2.00 ^b^
SBH	17.16 ± 0.80 ^ab^	94.22 ± 3.05 ^d^	16.35 ± 1.00 ^a^	75.55 ± 0.20 ^ab^

Note: SQA represents samples from SA in June–July; SAH represents samples from SA in August–September; SBQ represents samples from SB in June–July; and SBH represents samples from SB in August–September. The data are represented by the mean ± SD, n = 3. Different letters in the table represent a significant difference in the alpha diversity indices at the *p* < 0.05 level.

## Data Availability

Sequence data were deposited at the National Microbiology Data Center Raw Omics database (accession number for the fungal composition: NMDC40043833–NMDC40043844 and NMDC40043863–NMDC40043874). All other data are provided in this article’s results section and Appendix A.

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
