# Peer review of "The Diverse Mycorrizal Morphology of Rhododendron dauricum, the Fungal Communities Structure and Dynamics from the Mycorrhizosphere"

_jof, 2024, doi:10.3390/jof10010065_

Round 1

Reviewer 1 Report

Comments and Suggestions for Authors

Dear authors,

The manuscript ID JOF-2786920 has scientific merit and addresses an important topic. Knowledge of mycorrhiza-forming fungal communities is crucial for understanding the mechanisms that govern the fungus-plant symbiotic relationship. The study highlights important results about the mycorrhizal community of Rhododendron dauricum.

Although the study is relevant in the scientific field in which it is inserted, there are some gaps that the authors need to fill to increase the quality of the manuscript.

Review comments are in the PDF file.

Reviewer 2 Report

Comments and Suggestions for Authors

This is an interesting and promising study on a type of mycorrhiza that is little documented in general, but very important for the development of unique plants in the world that persist in very particular edaphic and environmental conditions, which is why this type of mycorrhiza is crucial for the conservation of this type of plants (Family Ericaceae). The manuscript provides unique information on the diversity of soil fungi and ectendomycorrhiza-forming fungi associated with a characteristic plant species of the Ericaceae family. The data are very complete, the manuscript is worthy of publication and provides very relevant information biologically and in terms of environmental and biological conservation, and even as a reference for the production of inocula that favor the development of this economically important plant.

However, in its current version the manuscript has some aspects that the authors must improve to increase the quality of their data presentation and understand their experimental design and data analysis before the manuscript is accepted for publication. These aspects were pointed out by this reviewer throughout the text of the manuscript, which can be seen in the pdf file.

Several details need to be addressed in the methods, including variables mentioned in the results that are not described in the methods. Furthermore, it is necessary to improve the presentation of the results in statistical terms, as well as the discussion limited only to fungi and not to all microbial communities. I ask that these aspects be carefully visualized and analyzed throughout the text of the manuscript.

Round 2

Reviewer 1 Report

Comments and Suggestions for Authors

Not applicable.

Author Response

Thank you for your review.

Reviewer 2 Report

Comments and Suggestions for Authors

The authors have considerably improved their manuscript by taking into account all the comments made by this reviewer, in addition to other aspects that they also improved to better express their results and present the introduction.

I found some details in this second review, however these will be corrected quickly: 1) The botanical descriptors of the species are not in italics, 2) It is striking that N and not P is the nutrient that is linked to the structure. of the fungal community despite the fact that the level of P in the soil is even lower than that of K and N. Strengthening this finding in the discussion could be very relevant.

However, I consider that the manuscript is suitable for publication in J of Fungi
